# Smart Wearable Systems for the Remote Monitoring of Selected Vascular Disorders of the Lower Extremity: A Systematic Review

**DOI:** 10.3390/ijerph192215231

**Published:** 2022-11-18

**Authors:** Julio Souza, Sara Escadas, Isidora Baxevani, Daniel Rodrigues, Alberto Freitas

**Affiliations:** 1Department of Community Medicine, Information and Health Decision Sciences (MEDCIDS), Faculty of Medicine of the University of Porto, 4200-450 Porto, Portugal; 2Department of Materials Science and Technology, University of Crete, 700 13 Iraklio, Greece

**Keywords:** mHealth, personalized healthcare, intermittent claudication, leg ulcer, diabetic foot, remote monitoring, continuous monitoring, wearable devices

## Abstract

This systematic review aims at providing an overview of the state of the art regarding smart wearable systems (SWS) applications to monitor the status of patients suffering from vascular disorders of the lower extremity. Peer-reviewed literature has been analyzed to identify employed data collection methods, system characteristics, and functionalities, and research challenges and limitations to be addressed. The Medline (PubMed) and SCOPUS databases were considered to search for publications describing SWS for remote or continuous monitoring of patients suffering from intermittent claudication, venous ulcers, and diabetic foot ulcers. Publications were first screened based on whether they describe an SWS applicable to the three selected vascular disorders of the lower extremity, including data processing and output to users. Information extracted from publications included targeted disease, clinical parameters to be measured and wearable devices used; system outputs to the user; system characteristics, including capabilities of remote or continuous monitoring or functionalities resulting from advanced data analyses, such as coaching, recommendations, or alerts; challenges and limitations reported; and research outputs. A total of 128 publications were considered in the full-text analysis, and 54 were finally included after eligibility criteria assessment by four independent reviewers. Our results were structured and discussed according to three main topics consisting of data collection, system functionalities, and limitations and challenges.

## 1. Introduction

The aging of populations has prompted increasing challenges for health systems around the world due to the higher demand for care and assistance, driven by the growing prevalence of chronic conditions, multimorbidity patterns, and disability [1]. On the other hand, health monitoring technologies have emerged as effective tools for the prevention, early detection, and management of chronic conditions [2]. The so-called smart wearable systems (SWS) are products that contain high-tech and portable components, including low-cost devices such as sensors, actuators, and communication components serving a specific purpose [1]. The main purpose of these devices is to monitor physiological parameters reflecting health status, physical activity, and other values associated with the psychological, cognitive, emotional, and mental state of patients, through sensors that transmit the collected data to a central system through communication modules [3]. These systems are able to measure a variety of parameters in real time and forward them via a wireless sensor network (WSN), either to a central connection node or directly to a medical center, where clinicians and caregivers can then manage the patient based on the transmitted data and implement interventions when needed, including in emergency situations [4]. Effective monitoring for longer periods than allowed during hospital stays or visits to the physicians’ offices is possible with these systems and can be used both inside and outside the home [5]. In the health domain, a variety of SWS to support independent living for the elderly, postoperative rehabilitation, and the analysis and improvement of individual health, technical or sportive abilities have been developed [6].

Wearable devices for health research are increasing as more options become accessible and affordable, and rates of clinical approval of respective regulatory bodies increase [2]. Amongst the numerous applications in the health care sector, the development of SWS in the field of vascular disorders affecting lower extremities such as diabetic foot ulcers (DFU), venous ulcers (VU), and intermittent claudication (IC) offer key opportunities, as these conditions can lead to considerable loss of mobility and quality of life, often requiring constant follow-up and hospital visits [7,8]. Standard treatment plans are defined based on patients’ condition and comprise pharmacological and physical activity interventions to manage disease progression and improve the overall condition. However, follow-ups are only done sporadically due to the difficulty to monitor the patients and their activities outside of healthcare facilities, opening key opportunities and gains with the development of wearable technologies that allows remote and continuous monitoring.

The pathophysiology of the diabetic foot is caused by neuropathy, arterial occlusive disease, and trauma with secondary infection. Peripheral neuropathy leads to intrinsic muscle atrophy, leading to functional anatomical changes in hammertoe formation and the development of high-pressure zones on the plantar surface of the metatarsal heads. Repetitive trauma with walking, together with decreased sensitivity, leads to skin damage and consequent displacement of the protective plantar fat pads, leading to ulceration [9]. Although DFU can purely be neuropathic, some forms are purely ischemic or a combination of both (neuroischemic), with the prevalence estimated at 35%, 15%, and 50% for each type, respectively [10]. Diabetic foot ulcers can be described as one of the most common and costly complications of diabetes, affecting around a quarter of diabetic patients, apart from being the leading cause of diabetes-related hospitalizations [11], accounting for one-fifth of the amputations of the lower limb among diabetic patients [12,13]. Moreover, the chances of DFU recurrence are estimated at 40% in the first year, increasing to almost 100% during a 10-year period [11]. Standard strategies for DFU prevention include a screening of high-risk insensate foot, regular foot care, adoption of standard therapeutic shoes and insoles to accommodate foot deformities and control high plantar pressures, as well as diabetic foot education [11]. Additionally, good patient adherence to the daily wearing of this therapeutic footwear is required to achieve treatment effectiveness [14]. In order to prevent the formation of ulcers on the patient's foot, parameters that can lead to their formation, such as temperature and pressure, must be monitored and controlled [15]. In this sense, the use of non-invasive tools is the best way to monitor various parameters such as foot temperature, pressure, and humidity [16].

Venous leg ulcers are the last and most severe stage of chronic venous disease according to the C5 and C6 classes within the Clinical Etiological Anatomical Pathophysiological (CEAP) classification, resulting from failures of the valves that connect the superficial and deeper veins, manifesting as superficial venous hypertension. Capillaries cannot withstand this high pressure for a long time, leading to a decrease in oxygen distribution and consequent ulcer development [17]. If timely and proper care is not given, the blood drips from the vein and the affected skin gets swollen and tight, leading to heavy pain. Treatment options available include the use of compression therapy, usually through compression bandages, compression stockings, or pneumatic compression devices. One of the potential solutions to this problem includes the use of an intelligent system that can manage the amount of pressure applied by measuring physiological variables [17].

Intermittent claudication (IC) is usually associated with peripheral arterial disease (PAD), which is a highly prevalent and debilitating condition affecting around 6% of patients above 60 years of age [18]. Lower extremity PAD is a chronic atherosclerotic occlusive condition causing insufficient blood flow to the lower extremities, resulting in walking pain in the lower limbs and impaired walking observed in IC individuals [19]. In this sense, IC can be defined as lower limb pain or discomfort after or during walking, which eases when the patient rests [20]. This condition results in a loss of function, loss of ability to walk, and a consequent decrease in quality of life [21]. The first line of treatment for this disease is supervised exercise therapy (SET), risk factor modification, and medication therapy [22]. However, poor patient compliance with supervised exercise programs, which can be as low as 34% [21], is a major constraint to IC treatment, as the benefit in walking distance is highly dependent on the frequency and maintenance of the exercises after the completion of the supervised programs. In recent years, approaches have been studied to increase the effectiveness of these exercises done at home, namely self-monitoring, and performance feedback. Given this, interest has grown in wearable activity monitors as part of the treatment for these patients suffering from IC [23].

Studies show that the continuous monitoring of physiological parameters of patients leads to an improvement in their health condition [7,24,25] and, as such, it has been sought the development of solutions that address the current difficulties in the healthcare of vascular disorders of the lower extremity. In this document, a systematic review of the literature related to SWS used for the treatment of VU, DFU, and IC is made. The main objective of this review is to provide an overview of the current state of the art regarding data collection methods, system characteristics, and capabilities, and current research challenges and limitations to be addressed.

## 2. Materials and Methods

### 2.1. Search Strategy

We conducted a literature search in databases of PubMed and SCOPUS in order to identify relevant studies describing SWS focusing on the monitoring of the previously described vascular disorders of the lower extremity. We searched from inception to the date of the last search, which was performed on 25 May 2022. To improve query recall, the search queries were defined considering the terms in titles, abstracts, and author-identified keywords. Queries fundamentally included terms to identify the intended system, referencing telemedicine, sensor, and device, among other relevant reference words to find articles of interest and adding wide-ranging terms for each one of the vascular disorders, which correspond to publications addressing diabetic foot ulcer, venous ulcers, and intermittent claudication. The final queries, after refining the query and performing sensitivity analysis, are reported below. PubMed Search Query: ((“sensors”[tw] OR “sensor”[tw] OR “Assistive technology”[tw] OR “ehealth”[tw] OR “mhealth”[tw] OR “Home healthcare”[tw] OR “Smart Mobile”[tw] OR “Implantable device”[tw] OR “Implantable devices”[tw] OR “Telecare”[tw] OR “Telehealth”[tw] OR “Telemedicine”[tw] OR “Portable”[tw] OR “Wearable”[tw] OR “Wearables”[tw] OR “smart health”[tw] OR “monitoring”[tw]) AND (“venous ulcer”[tw] OR “venous ulcers”[tw] OR “leg ulcer”[tw] OR “diabetic foot”[tw] OR “plantar pressure”[tw] OR “diabetic feet”[tw] OR “DFU”[tw] OR “foot ulcer”[tw] OR “foot ulcers”[tw] OR “claudication”[tw] OR “peripheral artery disease”[tw])). Scopus Search Query: ((“sensors” OR “sensor” OR “Assistive technology” OR “ehealth” OR “mhealth” OR “Home healthcare” OR “Smart Mobile” OR “Implantable device” OR “Implantable devices” OR “Telecare” OR “Telehealth” OR “Telemedicine” OR “Portable” OR “Wearable” OR “Wearables” OR “smart health” OR “monitoring”) AND (“venous ulcer” OR “venous ulcers” OR “leg ulcer” OR “diabetic foot” OR “plantar pressure” OR “diabetic feet” OR “DFU” OR “foot ulcer” OR “foot ulcers” OR “claudication” OR “peripheral artery disease”)).

### 2.2. Eligibility Criteria

Only studies written in English and published in peer-reviewed journals or conference proceedings were considered. Considering both screening and full-text analysis phases, documents were included if they (1) present a Smart Wearable System (SWS) applicable to any of the three selected vascular disorders of the lower extremity and that have been applied with real users; (2) include the use of a wearable or portable device; (3) and contain data processing and provide output to users. Exclusion criteria include non-English publications, documents with no abstract available, letters, editorials, comments, abstract-only publications, and study protocols. In the full-text analysis, an additional exclusion criterion was considered: other reviews or documents that comprise the analysis of several works and that are not original research publications.

Screening and full-text analysis phases were performed by four independent reviewers following the same strategy: the list of retrieved publications was divided into four batches with an equivalent number of publications and each batch was assessed by two different reviewers. Disagreements were resolved through consensus involving the assessment of all four reviewers. The complete list of included articles is presented in the References section [26,27,28,29,30,31,32,33,34,35,36,37,38,39,40,41,42,43,44,45,46,47,48,49,50,51,52,53,54,55,56,57,58,59,60,61,62,63,64,65,66,67,68,69,70,71,72,73,74,75,76,77,78,79].

### 2.3. Data Extraction and Quality Assessment

Regarding the data extraction phase, all articles that passed through a full-text analysis were divided into four groups and extraction occurred independently by each researcher. A random sample of 4 publications were extracted by all researchers to compare the similarity of the information extracted and to make the extraction process as homogeneous as possible. Information extracted from publications is reported in Table 1. Additionally, the same reviewers independently applied a quality checklist based on the STARE-HI model [80], in order to evaluate the quality of the included publications.

### 2.4. Analysis

The included publications were independently reviewed by three researchers. Information extracted from publications included targeted disease; data collection (parameters to be measured, wearable devices); system objectives; system outputs to the user; data science approaches to infer and predict the patient’s state and/or to deliver coaching interventions, recommendations or alerts; challenges and limitations reported; and authors’ conclusions. Analyses were carried out by three relevant thematic areas: (a) data collection through the remote monitoring of patients via wearable devices and clinical parameters; (b) system characteristics, including remote or continuous monitoring capabilities, as well as functionalities derived from basic or more advanced data processing, enabling patients and healthcare professionals with means to visualize, interpret and communicate the processed information, or receive coaching and personalized recommendations; and (c) current research challenges and limitations to be addressed.

## 3. Results

### 3.1. Study Selection Process

In Figure 1, it is possible to observe the detailed study selection process through the PRISMA flow diagram model1. When searching the two databases following the application of the final query, a total of 5872 documents were obtained, which resulted in 3585 publications after removing the duplicates between the two repositories (*n* = 2287 duplicates). In the title and abstract screening phase, a total of 3457 studies did not meet our inclusion criteria and were excluded. A total of 54 articles were finally included and analyzed.

Most of the publications focused on DFU problems (*n* = 31), followed by IC (*n* = 17) and venous ulcers (*n* = 6). Original research articles published in international journals comprised the majority of the included publications (*n* = 50), as only four documents were conference papers [52,59,60,78]. Additionally, the period encompassed in the included publications ranged between 1993 and 2022, with most publications occurring after 2012, reaching a peak in 2021 (*n* = 9) (Figure 2).

In Section 3.2, Section 3.3 and Section 3.4, the body of the literature was further summarized according to data collection methods, system characteristics, and current challenges and limitations to be addressed, respectively. Appendix A presents a description of each one of the included studies, namely targeted disease, sample size and mean age, study duration, objectives, and main research outputs.

### 3.2. Data Collection Methods

Data collection typically occurred from different types of sensors integrated into distinct wearable, implantable or portable devices. Table 2 and Table 3 summarize, respectively, the parameters collected and wearable technology employed by the different studies for each one of the targeted vascular disorders of the lower extremity.

In most publications (*n* = 31), data was still not being transmitted directly to medical centers or end-users, but rather raw measurements were downloaded from devices or visualized through software’s Graphical User Interface [59] to be analyzed by researchers or healthcare professionals. In some studies, processed data from body-worn sensors were typically transmitted to patients using mobile phones as the main information gateway [62,67,70,73,74,78]. In the work conducted by Schneider et al. (2019) [70], patients received at least two tailored text messages per week aiming at encouraging activity and strategic behavioral changes, whereas in Reyzelman et al. (2018) [67], Torreblanca González et al. [73] and Wang et al. (2021) [74] individuals were able to visualize images with foot temperature maps through their mobile phones, including alerts once foot temperature differences are detected [73]. Killeen et al. (2018) [55] describes an alert system that detects persistent localized temperature differences exceeding 1.75 °C between the left and right feet. Clinical staff can access these foot temperature maps derived from the scans through a secure online physician portal for triage [55]. Once an alert is prompted, a phone call is made to the patient to encourage offloading, reduce ambulation, correct feet elevation, and eventually clinical exams.

Overall, a wide range of strategies were adopted to transmit the information to providers and patients. Du et al. (2021) [38] described a system equipped with a triaxial accelerometer and gyroscope to collect temporal gait, balance and spatial parameters from diabetic foot patients, where digitized signals are transmitted in real-time via Bluetooth to a computer for analysis. Banks et al. (2020) [28] described that foot temperature maps (thermograms) derived from a mat were available to clinicians through an online portal for decision making, in which temperature data was automatically analyzed to detect temperature asymmetry between both feet. Similarly, in Gordon et al. (2020) [49], foot temperature data collected from the mat during approximately 20 s were encrypted and transmitted to the manufacturer for automated analysis according to the established clinical protocol. A foot mat described by Frykberg et al. (2017) [41] notifies the patients whenever the scan is complete and later transmits the scanned data wirelessly and securely to servers complying with the Health Insurance Portability and Accountability Act of 1996, which are managed by the own manufacturer. These data are then saved and processed in order to automatically detect foot temperature asymmetry [41].

In the works described by Gardner et al. (2014) [46] and Gardner et al. (2011) [47], wearable devices were used to facilitate the conduction of exercise programs, in which patients wore a commercial step watch during each exercise session and returned the device and a logbook to the research staff at the end of pre-defined weeks. Activity data were downloaded, and results were reviewed in order to provide feedback for the upcoming training month [46]. In Duscha et al. (2018) [39], physical activity data obtained from a Fitbit device were downloaded and summarized into reports to caregivers, who had access to the patients’ online accounts to provide technical support, physical activity monitoring, and motivation and feedback.

Regarding DFU applications, force or temperature sensors integrated into smart socks, insoles or footwear were described in 61.3% of the total DFU-related publications (*n* = 19) (see Table 3). These sensors were used, in particular, to determine plantar pressure and foot temperature at different points of the foot. These parameters were monitored to inform patients and healthcare professionals about the real status of the diabetic foot, in order to predict and prevent the occurrence of DFU, providing alerts of high plantar pressure or temperature whenever necessary. In the opposite direction, Gordon et al. (2020) [49], Banks et al. (2020) [28], Frykberg et al. (2017) [41], and Killeen et al. (2018) [55] opted for a portable rather wearable solution by developing foot mats for monitoring foot temperature and pressure to predict and prevent DFU. Other parameters explored in DFU-related studies included physical activity parameters (e.g., walked distance, steps count) [26,30,58,66,70], daily activity [66,70,76], and gait/kinematic characteristics [36,38,75], as well as humidity [62] and biomarkers such as tissue oxygen saturation (StO2) [57] (Table 2).

The monitoring of IC patients was mainly carried out with activity monitor sensors to quantify daily physical activity in terms of walked distance (*n* = 15), as well as the amount of time spent doing exercises, often analyzed by type and intensity of exercises or activity (*n* = 6) (see Table 2). Physical activity intensity was quantified using different methods. Armstrong et al. (2014) [26] collects the number of steps taken continuously over time, including the time of day each step occurred, whereas Bus et al. (2012) [30] considered only the number of steps per minute. Lott et al. (2005) [58] focused on the number of strides taken per day, defined as “heel-strike of one foot to heel-strike of the same foot for the next successive step”. Wrobel et al. (2014) [76] also measured and analyzed several parameters related to the number of strides, including stride velocity, length, and stride time, stance and double stance phases as a percentage of stride time, as well as gait speed variability. Schneider et al. (2019) [70] monitored physical activity intensity by computing the number of steps per day and the percentage of the time patients wore the provided smartwatch. Moreover, daily mobility was estimated using an algorithm that extracts information such as stops and trips from raw GPS trajectory data, resulting in the number of places visited per day [70]. Owings et al. (2009) [66] measured activity through an activity score calculated as total standing hours added to twice the number of total walking hours.

Wearable technology most frequently employed in these applications included pedometers and tri-axial accelerometers in devices placed at distinct body sites, such as ankle [32,33,39,43,44,45,46,47], hips [33,72] and wrist, through smartwatches or similar products [32,39]. A few studies also collected other physiological parameters, such as energy expenditure during physical activities expressed in terms of Metabolic Equivalents [56] and ankle/brachial index [43,44,64,72]. The most relevant outcomes for IC treatment, usually estimated from the data collected from the sensors, included peak walking time (PWT) [32,42,45,47,48], which is defined as the walking time at which ambulation cannot continue due to maximal pain, claudication onset time (COT) [32,42,46,47], defined as the walking time at which the individuals first experienced pain. Outcomes recorded to quantify IC severity included initial claudication distance (ICD) [44], which corresponds to the walking distance at which the patient experienced pain for the first time, and absolute claudication distance (ACD) [43,44], defined as the distance at which walking could not continue due to maximal pain. The main objective of these systems was to improve the walking capacity of IC patients and thus optimize the distance they can walk without pain.

Regarding venous ulcers, force sensors placed under bandaging were used to determine the interface pressure applied by compression products that are typically applied in the ulcer area, providing means to adjust or change bandaging in an adequate time [65]. The use of multi-electrode sensors placed in the dressing was also used to monitor ulcer progression through bioimpedance analysis [51,52]. Some projects focused on physical activity parameters by using footwear-based Bluetooth-enabled triaxial accelerometers paired with mobile phone applications in order to improve lower leg function and strengthen lower extremities in individuals with leg venous ulcers, also recurring to communication functionalities with clinicians [53,54]. Henricson et al. (2021) [50] employed a moisture sensor placed on the patient’s dressings in order to detect moisture concerning the absorbing capacity of the dressings.

### 3.3. Current Features of Wearable Systems

Generally, applications described in the peer-reviewed publications employed wearable devices to facilitate patient follow-up, without major data processing by Artificial Intelligence (AI) algorithms. Most projects for the targeted diseases address continuous monitoring through technology, but only nearly half of the DFU and VU-related publications described remote monitoring of patients, whereas this feature was present in only a minority of IC-related publications (Figure 3). A few DFU and VU-related publications, however, mention the provision of feedback to users, such as alerts and recommendations based upon the processing of the data collected from sensors (smart or coaching component), although this level of functionality has not been found for IC applications (Figure 3).

Wearable devices used in the context of IC were implemented to help to motivate the patients to achieve their daily step goals, as part of their treatment plans defined by clinicians. No coaching or recommendation systems have been described in the assessed publications (see Figure 3). In fact, most of the described IC applications included these devices as part of exercise programs or used them to promote patient education. Garner et al. (2014) [46] proposed the inclusion of a step watch monitor attached to the ankle in a supervised exercise program and compares changes in PWT and COT in comparison with regular exercise programs, without the use of wearable technologies. In this sense, patients wore the step activity monitor during each exercise session and returned the monitor and a logbook to the study staff at the end of pre-defined weeks, in which monitor data were downloaded and reviewed and feedback was provided for the upcoming month of training during a 15 min meeting. Duscha et al. (2018) [39] used an activity tracker provided by a commercial smartwatch paired with a mobile application to disseminate patient education and daily exercise prescriptions based on the number of steps per day. Physical activity data were summarized into reports to be presented to caregivers and the study staff, which in turn provided support to technical problems, monitored physical activity, and provided additional motivation and feedback during the study period [39]. On the hand, more advanced analytics has been described by Clarke et al. (2013) [34], who developed a method of event-based analysis that quantified the typical nature of walking bouts among IC patients. This method classified events into sedentary, standing, and individual steps, apart from computing the total number of upright events (defined as a combination of standing and walking events), and the event-based claudication index (EBCI), which is the ratio of walking to upright events. The output provided to users includes the classification of events and from this primary classification, continuous walking events can be estimated by combining individual step events [34].

DFU was the condition for which most coaching functionalities have been implemented. Killeen et al. (2018) [55] developed a system where the clinical staff is warned to call the patient for triage whenever persistent localized temperature differences exceeding 1.75 °C between the left and right foot are detected. The patient is further monitored more carefully for a minimum of 2 weeks until the asymmetry episodes are solved. Moreover, phone calls are made to provide offloading instructions, decrease ambulation, the elevation of feet, self-exam, and eventually clinical exam. Clinical staff can monitor the patient through foot temperature maps, or thermograms, derived from the scans through a secure online physician portal [55]. Torreblanca González et al. (2021) [73] used stepwise regression to predict when temperature measurements exceed a certain threshold so that a smart sock would send an alert to the telephone the patient in order to decrease ambulation. Chatwin et al. (2018) [31] implemented high-pressure alerts to the patients whenever plantar pressure is greater than 35 mmHg sustained for 95 to 100% of the time in a 15 min sampling window on the sensor. These alerts are transmitted from the individuals’ smartwatch to notify the patient and further encourage offloading.

Some enhanced functionalities comprising alerts and recommendations were found in half of the publications addressing venous ulcers. Henricson et al. (2021) [50] implemented an alert through a moisture sensor placed in the dressings of venous ulcer patients. This alert consists of a blue drop, denominated display activation, indicating a need to change the dressing. Patients were instructed to monitor the provided display and report whether a blue drop (display activation) appeared between dressing changes. At each dressing change, the health care professional checked the display for activation, weighed the dressing, and recorded the wound status. The level of the dressings' absorbing capacities was determined by weighing the dressings before and after use. Kelechi et al. (2020) [53] described a mHealth solution with a communication feature between patients and providers for the self-management of physical activity among subjects with leg venous ulcers, including a 6-week exercise program and automated educational and motivational messages to patients, as well as user reports.

### 3.4. Research Challenges and Limitations

In what concerns the limitations of the reviewed publications, most studies reported small sample sizes, with difficulty in generalizing the results. In nearly all studies, the lack of robustness of the control group is highlighted, having only veterans in the study or most of the study population was male. The fact that patients volunteered was also a commonly mentioned limitation, referred to as self-selection bias regarding the interest and adherence to exercises, potentially influencing the results. Moreover, several studies drew conclusions based on results obtained using proof of concept or were carried out over a very short period.

Less frequent limitations, but also reported, include lack of control over room temperature (in studies whose sensors measure this variable), sensors that record activity but do not account for the time that participants are resting or practicing sedentary activities, and results that cannot be generalized to all patients, since individuals in the sample considered in the study may be in more or less advanced stages of the disease.

Furthermore, barriers were also reported in terms of adherence to the use of sensors, and technical limitations in terms of the sensors used, such as the non-accounting of asymmetries in gait or measurement failures. Lauret et al. (2014) [56] pointed to the considerable number of excluded participants due to incorrectly worn devices and a high dropout rate as major drawbacks of the study.

As future challenges, recommendations to carry out more in-depth studies in the future, with larger samples and longer study times have been extensively highlighted.

### 3.5. Quality Assessment of the Included Articles

Concerning the quality assessment of the included publications, we employed the STARE-HI checklist. This list includes a comprehensive set of relevant principles for properly describing health informatics evaluations. The assumption is that when manuscripts submitted to health informatics journals and general medical journals adhere to these aspects, studies would be placed in a proper context and their validity and generalizability would be better judged, including the degree to which publications fit in the scope of meta-analyses in specific health informatics interventions [80].

Table 4 presents the number of publications complying with each item proposed for the STARE-HI checklist. Overall, included publications presented a high level of compliance with the STARE-HI items, as most items presented 70% or more compliance, apart from item “Results—Unexpected observations”, in which only 37% of publications reported and described either positive or negative aspects potentially influencing the results, and items that do not necessarily reflect the quality of the publication, such as the presence of explicit authors’ contributions (31.5% of the total publications), competing interests (59.3% of the total publications) and supporting material in appendices (18.5% of the total publications). All included studies clearly specified study questions and hypotheses, including permissions obtained to conduct the study, described the overall study design and the rationale for choosing it, described the outcome measures of interest along with definitions of key concepts, and presented results so that each research question was addressed (Table 4).

## 4. Discussion

Vascular disorders affecting lower extremities considerably affect mobility and quality of life, and they tend to become more prevalent as populations age. The management of these diseases relies on medical appointments where treatment plans are defined according to the patient’s condition, comprising pharmacological interventions and physical activities that the patient needs to perform to control and improve their health status. The evolution of information technologies in the health sector, such as monitoring sensors embedded in wearable devices, has opened a possibility for a shift towards personalized healthcare, based on the remote and continuous monitoring of patient parameters. The use of smart coaching techniques to guide and motivate patients to adopt healthy behaviors, and support healthcare professionals during decision-making, are potential gains that may be associated with the use of wearable technology.

Two main sub-sections characterize a complete wearable medical device [81]: the hardware subsector, comprising sensor selection and characterization, noise removal from the collected data, and communication with a data processing subsystem. The other sub-sector, the software, comprises data processing and decisions based upon the collected data, in which AI techniques can be used for representing, modeling, and reasoning with medical evidence and knowledge, potentially demonstrating human-like capabilities for diagnosis, early detection, prevention and medical care guidance [82]. Therefore, in addition to remote and continuous motoring, prediction and recommendations with the data collected through wearable devices may potentially enhance SWS capabilities [81]. The present systematic review aimed at summarizing the state-of-the-art and current developments in terms of SWS applied to three relevant vascular disorders of the lower extremity. We summarized aspects of data collection, system characteristics, as well as current challenges and limitations for the development of these systems.

Research on SWS applied to diseases has shown an increased trend in recent years (see Figure 1). This is partially attributed to the marked progress in wearable sensors developments, which in turn is linked to progresses in embedded systems and material science [83], as well as the technological framework provided by Internet of Things (IoT), which facilitates data collection from mobile and wearable devices, apart from enhancing computing and storage capabilities through state-of-the-art technologies such as cloud computing [84]. However, the development of SWS applied to health is still an emerging field that has only been growing in recent years. Several challenges still need to be addressed, namely more research on battery technology in order to achieve greater energy efficiency, more efforts on implementing clear use cases that provide timely and valuable feedback and recommendations and more efforts to provide evidence on the cost-effectiveness and real improvements on clinical care pathways and workflow through the use of these systems [83].

Most of the reviewed publications presents novel devices for research purposes and are at proof-of-concept stage. However, in recent years, a variety of commercial products have been introduced and a considerable number of publications described the reuse of such commercially available devices. For instance, devices dedicated to fitness monitoring, such as wrist or arm bands and smart watches, namely Fitbit, which can provide real-time activity data under minimum hardware and computing abilities, thereby constituting cost-effective solutions for monitoring patients suffering from vascular disorders of the lower extremity. Moreover, these devices are usually capable of performing many other functions related to the smart phone [84].

A substantial share of the examined studies described technologies that allow continuous monitoring, although the minority enabled remote access to patient data. This occurs because several projects are still in the proof-of-concept stage or at most mention a follow-up provided by healthcare professionals based upon the data measured by the monitoring sensors, such as integrating wearable devices into exercise programs for IC patients for monitoring physical activity, but without major data processing. Automated recommendations, coaching, or alert feedback are still scarce. Applications addressing DFU and VU described some functionalities implemented at this level, usually carried out by smartphones or sent directly from wearable devices, warning the patient on moments to reduce walking or when there is a risk of ulcer development, such as abnormal plantar pressure or temperature patterns, displaying useful information to patients and clinicians. Nevertheless, the assessed publications indicate that clinical knowledge and evidence on IC and DFU is robust enough to support future initiatives and thus improve the current technologies, especially with regard to the effects of clinical parameters, behaviors, and risk factors associated with the evolution of these conditions. VU remains an underexplored field, and there is a plethora of opportunities to develop innovative SWS in this field, namely, to guide compression therapy and help clinicians to check whether patients are properly following medical recommendations.

Finally, published research lacks clinical validation and evaluation of the impact on health outcomes and well-being. In fact, Macdonald et al. (2021) [85] reported that diabetic populations remain optimistic about the role of technology in supporting foot monitoring but delivering evidence of wearable device efficacy in preventing foot ulcerations would improve trust and the likelihood of future adoption. International organizations such as Food and Drug Administration (FDA) and the European Commission present guidelines that deal with or encompass wearable devices used for medical purposes [86,87]. Considering that these devices work on multiple communication protocols, it is critical to establish their safety for human use before large scale implementation, thereby requiring the conduction of extensive clinical trials.

### Future Needs

There is a rising need for sustainable healthcare, with personalized treatment and management of patients, with increased proactivity of individuals regarding their own health condition. To address these needs by means of SWS and achieve a desirable efficacy, a robust infrastructure needs to be implemented for large-scale deployment of wearable devices integrated to conventional healthcare facilities. This integration requires challenges to be overcome at both clinical and operational levels. In terms of hardware, wearable devices may not be easily maintained, and they are affected by battery issues [84]. In terms of software, usable solutions must be provided to end-users, in which the presented information should ideally be sufficient, readable and detailed enough for providers to make proper decision-making and for patients to improve the activity levels and adopt healthy behaviors. Data types and volume collected via wearable devices have grown beyond the processing capabilities of regular data processing techniques [88]. In this sense, preprocessing methods such as noise removal, feature extraction and peak detection are critical for reducing the volume of data at the source [89]. Map-reduce tools also allow the efficient processing of large volumes of data [90]. Additionally, 5G technology provides ways to reduce latency, power and traffic demands to central communication nodes, which is useful for supporting scenarios where multiple devices are integrated and communicating with each other in the cloud [91]. Finally, ensuring data safety and confidentiality and complying with legislative guidelines often set by different institutions can be a major constraint for implementing these systems on a large scale.

From a clinical point of view, as future challenges, we highlight the recommendations to carry out more in-depth studies in the future, with larger samples and longer study times. Future work perspectives presented by the reviewed articles included mostly the refinement of current technologies and further evaluation and clinical validation of existing projects.

## 5. Conclusions

We reviewed the current state-of-the art on SWS for monitoring vascular disorders affecting lower extremities, summarizing aspects of data collection, existing wearable devices and sensing solutions, system features, limitations, challenges and future needs. Although this is still and emerging field that have been growing just recently, important achievements have been made, mostly due to progresses in areas such as wearable sensors, embedded systems, material science, IoT communication and cloud computing. Most publications presented novel devices enabling continuous monitoring, but systems allowing remote monitoring and smart coaching and recommendations are still less common in the field of vascular disorders affecting the lower limb. Several projects concerning IC employed novel or commercially available sensing devices to perform follow-up and manage exercise programs based upon the measured activity data, without using or implementing advanced analytics. Recommendations or coaching feedback, whenever available, were usually carried out by smartphone applications, warning diabetic foot patients when they should offload or whenever abnormal temperature patterns indicating elevated risk of ulcer development are detected. Clinical knowledge and evidence regarding the association between physiological and motion parameters and disease progression are robust for DFU and IC, proving evidence and clinical knowledge that can be used as basis for further developing recommendations and coaching level functionalities. Research on VU monitoring is still scare, and the available projects focused mostly on monitoring pressure under compression apparatus and wound moisture. Overall, most projects lack clinical validation, as works are mostly at proof-of-concept stage, focusing on pilots or small-scale preliminary studies. Future work tends to focus on the refinement of existing technologies and further clinical validation, providing evidence on the efficacy of such devices in providing actual gains in health outcomes, which in turn is critical to increase general acceptance by patients and clinicians.

## Figures and Tables

**Figure 1 ijerph-19-15231-f001:**
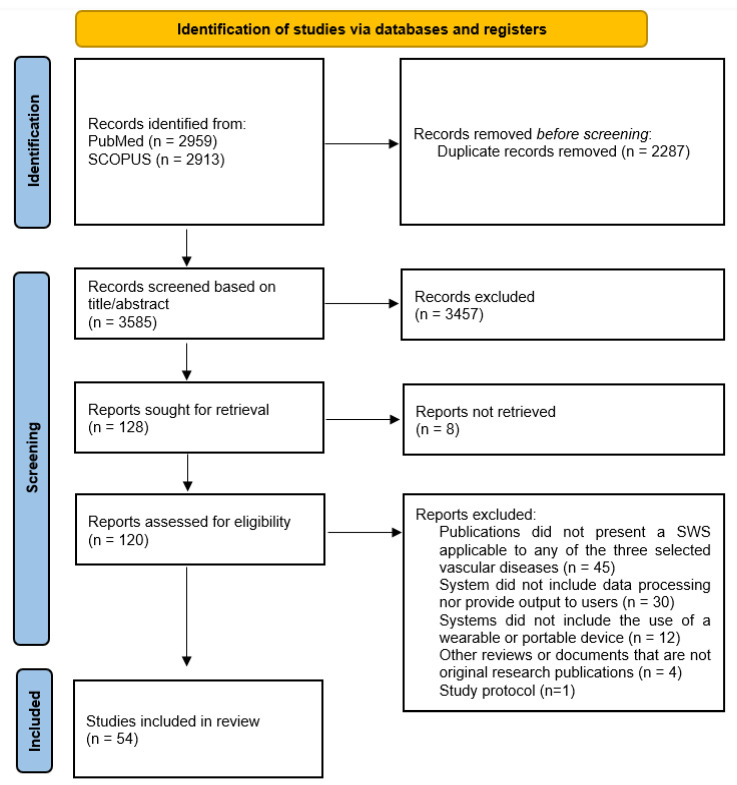
Flow diagram of the study selection process.

**Figure 2 ijerph-19-15231-f002:**
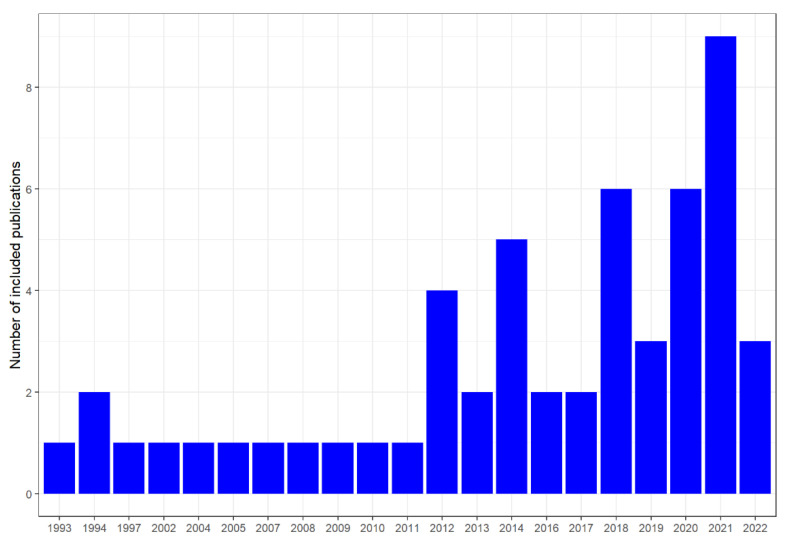
Number of included publications over time.

**Figure 3 ijerph-19-15231-f003:**
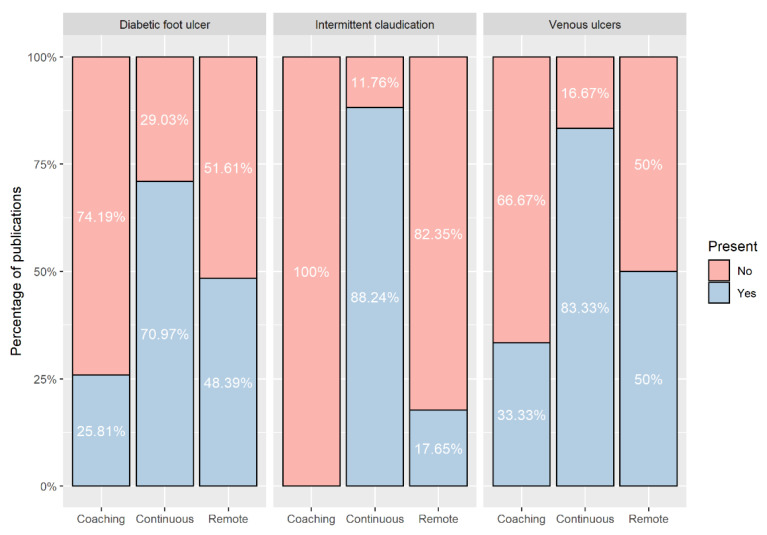
Share of number of publications presenting coaching (including alerts and recommendations), continuous and remote monitoring capabilities, per disease.

**Table 1 ijerph-19-15231-t001:** List of variables considered for data extraction.

Variables
Disease
Sensor and devices
Time frame of the study
Sample size
Participants age
Variables/Parameters
Feedback/Output to the user
Does it allow Remote monitoring? (Yes/No)
Does it allow Continuous monitoring? (Yes/No)
Is there any functionality of Coaching/Recommendations/Alerts (Yes/No)
Challenges and Limitations

**Table 2 ijerph-19-15231-t002:** Clinical parameters measured across the included publications, per disease.

Disease	Clinical Parameters
Intermittent claudication (Peripheral Arterial Disease)	Physical activity measurement in terms of walked distance (e.g., walking distance, steps count, the highest number of steps walked in one period, exercise adherence, walking velocity cadence) [27,33,34,35,39,40,42,43,44,45,47,48,56,64,72] Gait/kinematic characteristics (e.g., gait acceleration, swing phase, stance time, single-support time, double-support time, the base of support, stride length, and velocity) [33,42] Event detection (e.g., upright events, sedentary, walking, sitting, standing) [34] Physical activity measurement in terms of Metabolic equivalents (METs) [56] Time or percentage of time spent in different daily activities and exercise intensities/wear time [35,39,43,44,45,56] Ankle/brachial index [43,44,64,72] Claudication onset time (COT) [32,42,46,47] Peak walking time (PWT) [32,42,45,46,47] Number of stops [48] Number of stops induced by pain [32] Initial claudication distance (ICD) [44] Absolute claudication distance (ACD) [43,44] Maximum walking distance (MWD, pain-free) [35,64] Biomarkers (e.g., Oxygen saturation (St02), oxygen intake, Ischemic window, high-sensitive C-protein levels) [44,45,46]
Venous ulcers	Interface Pressure with bandage [65] Physical activity parameters (e.g., exercise adherence, exercise intensity, accelerations of the lower limb movement) [53,54] Moisture [50] Bioimpedance measurements (Skin bioimpedance, Wound status index (WSI)) [51,52] Gait/kinematic characteristics (e.g., range of motion, strength) [53]
Diabetic foot ulcers	Plantar/foot pressure (e.g., barefoot plantar pressure, in-shoe plantar pressure) [29,31,36,37,58,59,61,62,63,66,69,74,79] Foot Temperature [28,30,41,49,55,60,62,67,68,71,73,76,77,78] Foot deformity (e.g., hammer toes, claw toes, hallux valgus, bunions, prominent metatarsal heads, midfoot, or other prominences) [66] Physical activity measurement in terms of walked distance (e.g., steps count over time) [26,30,58,66,70] Activity profile (e.g., hours of sleeping, sitting, standing, walking, housework, wear time, attendance to supervised exercises) [66,70,76] Gait/kinematic characteristics (e.g., acceleration, angular velocity, range of motion of the center of mass, stride length and velocity, double-limb support) [36,38,75] Shear force [61] Biomarkers (e.g., ankle-brachial index (ABI), Tissue blood volume (HbT), oxyhemoglobin (HbO2), deoxyhemoglobin (Hb), and tissue oxygen saturation (StO2)) [57] Humidity [62]

**Table 3 ijerph-19-15231-t003:** Wearable and portable technology are described across the included publications, per disease.

Disease	Wearable/Portable Technology
Intermittent claudication (Peripheral Arterial Disease)	Smartwatch [32,39] Wireless sensor nodes at the ankle and hip [33] Activity monitor sensors attached to the ankle [43,44,45,46,47] Accelerometer-based device at the anterior part of the right mid-thigh [34] Accelerometer-based devices placed on opposite hips [72] Accelerometer-based digital wristwatch-sized device [32] Pedometer-based devices placed on opposite hips [72] Pedometer built-in mobile phone [27] Pedometer device, unspecified body site [64] Tri-axial accelerometer attached to a belt [40,56] Global Positioning System (GPS) receiver device [32,50] Sensewear (R) Mini device (Bodymedia) on the right upper arm [35] Smart mat [42]
Venous ulcers	Pressure sensors placed under compression apparatus [65] Moisture sensor placed in the dressing [50] Multi-electrode sensors placed in the dressing [51,52] Footwear-based Bluetooth-enabled triaxial accelerometer affixed [53,54]
Diabetic foot ulcers	Smart socks [67,68,70,73] Insoles equipped with pressure or temperature sensors [29,37,58,61,62,66,69,76,77,79] Footwear-attached pressure sensors [31,59,63,66,74] In-shoe motion sensors [32,75] Platform equipped with pressure sensors [66] Foot mat [28,41,49,55] Waist-mounted accelerometer/pedometer [26] Cast walkers [29,36] Triaxial accelerometer and gyroscope placed at the tibias of both lower limbs, and in front of the patient’s lower leg and umbilical plane [38] Medical infrared thermography (IRT) device [60] Smartwatch [70] Ambient temperature sensor [78] Wireless wearable near-infrared spectroscopy (NIRS) device designed to make contact with the morbid limb at the dorsal foot [57]

**Table 4 ijerph-19-15231-t004:** Number of publications complying with each item proposed for the STARE-HI checklist.

Item	Description	Number of Publications Complying with the Item (%)
Title	The title should give a clear indication of the type of evaluated system and the study question as well as the study design.	45 (83.3)
Abstract	The abstract must clearly describe the objective, setting, participants, measures, study design, major results, and conclusions.	52 (96.3)
Keywords	Among the keywords should be “evaluation” and keywords describing the type of system evaluated, the setting, outcome measures, and study design.	32 (59.3)
Introduction—Scientific Background	Description of what is already known about the (type of) intervention that is the object of study, what are still open research questions, and why there is a need to answer them.	49 (90.7)
Introduction—Rationale for the study	Short description of the motivation for the study; stakeholders and actors.	53 (98.1)
Introduction—Objectives of the study	The specific study questions and hypotheses, accompanied by permissions obtained in relation to the study.	54 (100)
Study Context—Organizational setting	The name, location, and kind of health care facility and involved departments.	38 (70.4)
Study Context—System details and system in use	A description that enables the reader to understand how the system works (or is intended to work) and its phase in the system’s life cycle.	52 (96.3)
Methods—Study design	The overall study design and the arguments for choosing it.	54 (100)
Methods—Theoretical background	Theories—with appropriate references—on which the study is based and that guided the selection of the measurement instruments used.	53 (98.1)
Methods—Participants	Methods of selection of participating users, patients, units, hospitals, etc., including if applicable inclusion and exclusion criteria.	50 (92.6)
Methods—Study flow	Details on the date of beginning and end of the overall study and any study periods with clear descriptions of intervention.	47 (87)
Methods—Outcome measures or evaluation criteria	Description of outcome measures used or other evaluation variables of interest together with definitions of key concepts.	54 (100)
Methods—Methods for data acquisition and measurement	Provide sufficient detail on data acquisition and measurement such that others are able to assess the appropriateness and any limitations, as well as to be able to replicate the measurement procedures of the study.	52 (96.3)
Methods—Methods for data analysis	For quantitative data, state which statistical techniques were used for analysis. For qualitative data, indicate the analysis methods in detail. For all data analysis methods, indicate any software product used.	46 (85.2)
Results—Demographic and other study coverage data	Baseline demographic data and clinical characteristics of study participants (users, patients, and units) and the study.	47 (87)
Results—Unexpected events during the study	Any unforeseen events that may have influenced the study results or outcome.	24 (44.4)
Results—Study findings and outcome data	Presenting the results of the study for each study question, for each outcome variable, and evaluation criterion.	54 (100)
Results—Unexpected observations	Any unintended (positive or negative) side-effects of the system that were not the focus of the study.	20 (37)
Discussion—Answers to study questions	A discussion of the answers identified versus the questions posed for the study.	53 (98.1)
Discussion—Strengths and weaknesses of the study	Critical discussion of the methods used.	46 (85.2)
Discussion—Results in relation to other studies	Make clear what exactly is novel about the obtained results.	52 (96.3)
Discussion—Meaning and generalizability of the study	The implication of the study findings, for the various stakeholders within the study and beyond.	52 (96.3)
Discussion—Unanswered and new questions	Future research needs and opportunities.	42 (77.8)
Conclusion	Summary of the main findings, including the impact of the findings and how they relate to the big picture provided.	53 (98.1)
Authors’ contribution	Explicit description of the contributions of the authors to make sure that each author qualifies for authorship.	17 (31.5)
Competing interests	A statement of the interest, financial or otherwise, the authors may have with respect to the outcome of the study.	32 (59.3)
Acknowledgment	Acknowledgments of any financial or other support.	39 (72.2)
References	All references are needed for the argumentation.	50 (92.6)
Appendices	Any supporting material, such as detailed descriptions of methods/tools (e.g., a questionnaire), specific data analysis techniques, and detailed study results.	10 (18.5)

## Data Availability

Data supporting the results can be found in Appendix A.

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
