# Peer review of "Smart Wearable Systems for the Remote Monitoring of Selected Vascular Disorders of the Lower Extremity: A Systematic Review"

_ijerph, 2022, doi:10.3390/ijerph192215231_

Round 1
Reviewer 1 Report
Within this manuscript Sousa et al. aim at providing a comprehensive review about the contribution of smart wearable systems (SWS) applications in monitoring the health of patients suffering from cardiovascular diseases.
They focus on three different diseases: diabetic foot ulcers, venous ulcers and intermittent claudication.
I think the review is well written and easy to follow. The methods, analysis and results are properly described together with the inclusion/exclusion of the literature considered. Overall, this work will provide useful information to the people of this field.
Author Response
Within this manuscript Sousa et al. aim at providing a comprehensive review about the contribution of smart wearable systems (SWS) applications in monitoring the health of patients suffering from cardiovascular diseases.
They focus on three different diseases: diabetic foot ulcers, venous ulcers and intermittent claudication.
I think the review is well written and easy to follow. The methods, analysis and results are properly described together with the inclusion/exclusion of the literature considered. Overall, this work will provide useful information to the people of this field.
Answer: Thank you very much for your comments and feedback.

Reviewer 2 Report
Souza, et al performed a review of smart wearable devices that is a timely article. They have pulled together a variety of manuscripts, which have assessed a range of devices. Overall, the review is well conceived with appropriate search and evaluation criteria. The figures and tables are illustrative and help the reader understand the context. The text is reasonably well written. There are several major and minor comments regarding the work.
1. Claudation and venous ulcers can be broadly characterized as venous disorders, but diabetic ulcers are not as strictly vascular. Rather diabetic ulcers are more related to neuropathology issues, with hemorrhage occurring later. It would be worthwhile to add a bit more detail regarding the etiology of disease to provide the reader a bit more background.
2. The title indicates that there will be a focus on vascular diseases. Most often vascular biologists would think of issues related to hypertension. Given this, it is worth considering modifying the title to better reflect the disorders that this article addresses.
3. There are multiple minor grammatical errors throughout the manuscript that detract from the readability. Please read through and edit accordingly. A few examples are on lines 57, 101, lines 261-263, line 315
4. Line 51: Make the order of the list of diseases match the order of the paragraphs and sections.
5. Line 96: Need to expand on the idea of continuous monitoring. There is no reference to what is being monitored...Even something as simple as “continuous monitoring of physiological parameters” would be helpful.
6. Figure 2 is good as it shows the trajectory of research in this field, however it would be worthwhile to discuss how this is still an emerging area of research that has only been growing for a few years.
7. Line 192: Some discussion is warranted regarding whether devices are commercially available or only for research purposes?? It seems that there is a mixture of both.
8. Table 2: Need to have additional demarcation between each disease section. The "space" that is provided between the various sections isn’t quite enough delineation.
9. Line 199: What format is the data sent to the phone. Are these all bluetooth devices or do they use ANT+ or other technologies. What are the battery sources. Are they rechargeable or are they disposable batteries. Any other consumables associated with them (i.e. stick on electrodes??)
10. Elsewhere there is discussion of transmission of information to medical providers. How is that achieved. Is it using an open link through wife or cellular data allowing for continuous remote monitoring? Is that phone-based communication? Are the activities uploaded following the end of the workout for subsequent review such as what is done with commercial devices from Garmin, Fitbit and others from cycling and running industries??
11. Line 216: How was intensity quantified?? Were they using heart rate-based tracking, speed, are they using GPS technologies for tracking individuals and gaining more granular information regarding distance, terrain, and activity time??
12. Section 3.3: Should consider a discussion of the data formats. Are the data in a proprietary format or can the data be accessed in a CSV or other format that allows to experimenter to perform post-hoc analysis using unique algorithms, and ability to provide written feedback. Can the data be synchronized with cloud-based packages allowing both the patient and providers to access the data? Can this be interfaced with platforms that are routinely used to provide training prescriptions (e.g. Trainingpeaks)?
13. Line 335: section 3.4 - Line 335: Be more specific regarding the statement “Quality Assessment”. This section is a quality assessment of the articles vs assessment of the devices themselves.
14. Line 356: Table 4 title. This seems to be placeholder text. Revise accordingly.
15. Section 4: In the discussion include a subsection specifically titled "Future Needs" or something like that. Sufficient content is written regarding what the industry needs but it should be brought together in a concise way. Subsections could also be made highlighting hardware and software elements detailing what the ideal elements are.
16. Line 401: One thing to consider is with regards to clinical validation. Importantly, it may be warranted to discuss EU and FDA approval for these as medical devices as opposed to using them simply as consumer technologies.
Author Response
- Claudication and venous ulcers can be broadly characterized as venous disorders, but diabetic ulcers are not as strictly vascular. Rather diabetic ulcers are more related to neuropathology issues, with hemorrhage occurring later. It would be worthwhile to add a bit more detail regarding the etiology of disease to provide the reader a bit more background.
Answer: Thank you for your insightful comment. Although the three diseases can be a result of vascular insufficiency, we have now clarified that DFU can be purely neuropathic, purely ischemic, or a combination of both (neuroischemic). Moreover, more details regarding the etiology of the diseases have been added in the Introduction section:
“The pathophysiology of the diabetic foot is caused by neuropathy, arterial occlusive disease, and trauma with secondary infection. Peripheral neuropathy leads to intrinsic muscle atrophy, leading to functional anatomical changes in hammertoe formation and the development of high-pressure zones on the plantar surface of the metatarsal heads. Repetitive trauma with walking, together with decreased sensitivity, leads to skin damage and consequent displacement of the protective plantar fat pads, leading to ulceration [9]. Although DFU can purely be neuropathic, some forms are purely ischemic or a combination of both (neuroischemic), with the prevalence estimated at 35%, 15%, and 50% for each type in 2011, respectively [10].”
“Venous leg ulcers are the last and most severe stage of chronic venous disease according to the C5 and C6 classes within the Clinical Etiological Anatomical Pathophysiology-logical (CEAP) classification, resulting from failures of the valves that connect the superficial and deeper veins, manifesting as superficial venous hypertension. Capillaries cannot withstand this high pressure for a long time, leading to a decrease in oxygen distribution and consequent ulcer development [17]. If timely and proper care is not given, the blood drips from the vein and the affected skin gets swollen and tight, leading to heavy pain. Treatment options available include the use of compression therapy, usually through compression bandages, compression stockings, or pneumatic compression devices. One of the potential solutions to this problem includes the use of an intelligent system that can manage the amount of pressure applied by measuring physiological variables [17].
“Intermittent claudication (IC) is usually associated with peripheral arterial disease (PAD), which is a highly prevalent and debilitating condition affecting around 6% of patients above 60 years old of age [18]. Lower extremity PAD is a chronic atherosclerotic occlusive condition causing insufficient blood flow to the lower extremities, resulting in walking pain in the lower limbs and impair walking observed in IC individuals [19]. In this sense, IC can be defined as lower limb pain or discomfort after or during walking, which eases when the patient rests [20]. This condition results in a loss of function, loss of ability to walk, and a consequent decrease in quality of life [21]. The first line of treatment for this disease is supervised exercise therapy (SET), risk factor modification, and medication therapy [22].”
- The title indicates that there will be a focus on vascular diseases. Most often vascular biologists would think of issues related to hypertension. Given this, it is worth considering modifying the title to better reflect the disorders that this article addresses.
Answer: Thank you for your suggestion. In order to better specify the targeted diseases and scope down the application areas, we have now changed the title to “Smart wearable systems for the remote monitoring of selected vascular disorders of the lower extremity: a systematic review”. Changes in the manuscript have also been made accordingly, in order to avoid the use of the broader term “vascular diseases", using “vascular disorders of the lower extremity” instead.
- There are multiple minor grammatical errors throughout the manuscript that detract from the readability. Please read through and edit accordingly. A few examples are on lines 57, 101, lines 261-263, line 315
Answer: Changed accordingly.
- Line 51: Make the order of the list of diseases match the order of the paragraphs and sections.
Answer: Changed accordingly.
- Line 96: Need to expand on the idea of continuous monitoring. There is no reference to what is being monitored...Even something as simple as “continuous monitoring of physiological parameters” would be helpful.
Answer: We have now indicated the continuous monitoring of physiological parameters, which better specifies the key objective of the targeted systems.
- Figure 2 is good as it shows the trajectory of research in this field, however it would be worthwhile to discuss how this is still an emerging area of research that has only been growing for a few years.
Answer: Thank you for your relevant comment. To address this query, we have now discussed possible factors related to the recent research growing and why this field is still an emerging trend. The following paragraph has been added to the Discussion section:
“Research on SWS applied to diseases has showed an increased trend in recent years (see Figure 1). This is partially attributed to the marked progress in wearable sensors developments, which in turn is linked to progresses in embedded systems and material science [83], as well as the technological framework provided by Internet of Things (IoT), which facilitates data collection from mobile and wearable devices, apart from enhancing computing and storage capabilities through state-of-the-art technologies such as cloud computing [84]. However, the development of SWS applied to health is still an emerging field that has only been growing in the last years. Several challenges still need to be addressed, namely more research on battery technology in order to achieve greater energy efficiency, more efforts on implementing clear use cases that provide timely and valuable feedback and recommendations and more efforts to pro-vide evidence on the cost-effectiveness and real improvements on clinical care path-ways and workflow through the use of these systems [83].”
- Line 192: Some discussion is warranted regarding whether devices are commercially available or only for research purposes?? It seems that there is a mixture of both.
Answer: In fact, we have observed that there is a mixture of both, commercially available devices that were later used for research purposes, with FitBit being a recurrent example, and several technologies that were developed by research purposes. We have now included in the Discussion section:
“Most of the reviewed publications presents novel devices for research purposes and are at proof-of-concept stage. However, in recent years, a variety of commercial prod-ucts have been introduced and a considerable number of publications described the reuse of such commercially available devices. Devices dedicated to fitness monitoring, such as wrist or arm bands and smart watches, namely Fitbit, which can provide re-al-time activity data under minimum hardware and computing abilities, thereby con-stituting cost-effective solutions for monitoring patients suffering from vascular dis-orders of the lower extremity. Moreover, these devices are usually capable of perform-ing many other functions related to the smart phone [84].”
- Table 2: Need to have additional demarcation between each disease section. The "space" that is provided between the various sections isn’t quite enough delineation.
Answer: Changed accordingly.
- Line 199: What format is the data sent to the phone. Are these all bluetooth devices or do they use ANT+ or other technologies. What are the battery sources. Are they rechargeable or are they disposable batteries. Any other consumables associated with them (i.e., stick on electrodes??)
Answer: Although we understand the importance of providing this information to enrich the review, detailed information on the technology is used to transmit the data was not present in most of the articles, as well as battery sources. Some information, however, can now be found in the Results section, as we have included several examples on how data was transmitted, including those recurring to technologies such as Bluetooth.
- Elsewhere there is discussion of transmission of information to medical providers. How is that achieved. Is it using an open link through wife or cellular data allowing for continuous remote monitoring? Is that phone-based communication? Are the activities uploaded following the end of the workout for subsequent review such as what is done with commercial devices from Garmin, Fitbit and others from cycling and running industries??
Answer: Thank you for your comment. Have now included in the Results more information and examples regarding data transmission to end-users, in attempt to describe the variety of strategies that have been implemented:
“In most publications (n=31), data was still not being transmitted directly to medical centers or end-users, but rather raw measurements were downloaded from devices or visualized through software’s Graphical User Interface [59] to be analyzed by re-searchers or healthcare professionals. In some studies, processed data from body-worn sensors were typically transmitted to patients using mobile phones as the main information gateway [62,67,70, 73, 74,78]. In the work conducted by Schneider et al. (2019) [70], patients received at least two tailored text messages per week aiming at encouraging activity and strategic behavioral changes, whereas in Reyzelman et al. (2018) [67], Torreblanca González et al. [73] and Wang et al. (2021) [74] individuals were able to visualize images with foot temperature maps through their mobile phones, including alerts once foot temperature differences are detected [73]. Killeen et al. (2018) [55] describes an alert system that detects persistent localized temperature differences exceeding 1.75ËšC between the left and right feet. Clinical staff can access these foot temperature maps derived from the scans through a secure online physician portal for triage [55]. Once an alert is prompted, a phone call is made to the patient to encourage offloading, reduce ambulation, correct feet elevation, and eventually clinical exams.
Overall, a wide range of strategies were adopted to transmit the information to providers and patients. Du et al. (2021) [38] described a system equipped with a triaxial accelerometer and gyroscope to collect temporal gait, balance and spatial parameters from diabetic foot patients, where digitized signals are transmitted in real-time via Bluetooth to a computer for analysis. Banks et al. (2020) [28] described that foot temperature maps (thermograms) derived from a mat were available to clinicians through an online portal for decision making, in which temperature data was automatically analyzed to detect temperature asymmetry between both feet. Similarly, in Gordon et al. (2020) [49], foot temperature data collected from the mat during approximately 20 seconds were encrypted and transmitted to the manufacturer for automated analysis according to the established clinical protocol. A foot mat described by Frykberg et al. (2017) [41] notifies the patients whenever the scan is complete and later transmits the scanned data wirelessly and securely to servers complying with the Health Insurance Portability and Accountability Act of 1996, which are managed by the own manufacturer. These data are then saved and processed in order to automatically detect foot temperature asymmetry [41].
In the works described by Gardner et al (2014) [46] and Gardner et al (2011) [47], wearable devices were used to facilitate the conduction of exercise programs, in which patients wore a commercial step watch during each exercise session and returned the device and a logbook to the research staff at the end of pre-defined weeks. Activity data were downloaded, and results were reviewed in order to provide feedback for the upcoming training month [46]. In Duscha et al. (2018) [39], physical activity data obtained from a Fitbit device were downloaded and summarized into reports to caregivers, who had access to the patients’ online accounts to provide technical support, physical activity monitoring, and motivation and feedback.”
- Line 216: How was intensity quantified?? Were they using heart rate-based tracking, speed, are they using GPS technologies for tracking individuals and gaining more granular information regarding distance, terrain, and activity time??
Answer: We have now included more information on how activity intensity was quantified in the Results section:
“Physical activity intensity was quantified using different methods. Armstrong et al. (2014) [26] collects the number of steps taken continuously over time, including the time of day each step occurred, whereas Bus et al. (2012) [30] considered only the number of steps per minute. Lott et al. (2005) [58] focused on the number of strides taken per day, defined as “heel-strike of one foot to heel-strike of the same foot for the next successive step”. Wrobel et al. (2014) [76] also measured and analyzed several parameters related to the number of strides, including stride velocity, length, and stride time, stance and double stance phases as a percentage of stride time, as well as gait speed variability. Schneider et al. (2019) [70] monitored physical activity intensity be computing the number of steps per day and the percentage of the time patients wore the provided smartwatch. Moreover, daily mobility was estimated using an algorithm that extracts information such as stops and trips from raw GPS trajectory data, resulting in the number of places visited per day [70]. Owings et al. (2009) [66] measured activity through an activity score calculated as total standing hours added to twice the number of total walking hours.”
- Section 3.3: Should consider a discussion of the data formats. Are the data in a proprietary format or can the data be accessed in a CSV or other format that allows to experimenter to perform post-hoc analysis using unique algorithms, and ability to provide written feedback. Can the data be synchronized with cloud-based packages allowing both the patient and providers to access the data? Can this be interfaced with platforms that are routinely used to provide training prescriptions (e.g. Trainingpeaks)?
Answer: Thank you very much for your insightful comment and suggestion. However, we opted for not including information because a minority of the articles actually describe data format in such level, so the scope of the review focused mostly on the parameters, devices and how the collected data is typically transmitted for analysis or feedback.
- Line 335: section 3.4 - Line 335: Be more specific regarding the statement “Quality Assessment”. This section is a quality assessment of the articles vs assessment of the devices themselves.
Answer: We have now clarified that this is a quality assessment of the articles by changing the title of the subsection.
- Line 356: Table 4 title. This seems to be placeholder text. Revise accordingly.
Answer: Changed accordingly.
- Section 4: In the discussion include a subsection specifically titled "Future Needs" or something like that. Sufficient content is written regarding what the industry needs but it should be brought together in a concise way. Subsections could also be made highlighting hardware and software elements detailing what the ideal elements are.
Answer: Thank you for your insightful comment. We have now provided a more detailed discussion on future needs, highlighting some hardware and software aspects to be addressed:
“There is a rising need for sustainable healthcare, with personalized treatment and management of patients, with increased proactivity of individuals regarding their own health condition. To address these needs by means of SWS and achieve a desirable efficacy, a robust infrastructure needs to be implemented for large-scale deployment of wearable devices integrated to conventional healthcare facilities. This integration re-quires challenges to be overcome at both, clinical and operational levels. In terms of hardware, wearable devices may not be easily maintained, and they are affected by battery issues [84]. In terms of software, usable solutions must be provided to end-users, in which the presented information should ideally be sufficient, readable and detailed enough for providers to make proper decision-making and for patients to improve the activity levels and adopt healthy behaviors. Data types and volume collected via wearable devices have grown beyond the processing capabilities of regular data processing techniques [88]. In this sense, preprocessing methods such as noise removal, feature extraction and peak detection are critical for reducing the volume of data at the source [89]. Map-reduce tools also allow efficient processing of large volumes of data [90]. Additionally, 5G technology provides ways to reduce latency, power and traffic demands to central communication nodes, being useful to support scenarios where multiple devices are integrated and communicating with each other in the cloud [91]. Finally, ensuring data safety and confidentiality and complying with legislative guidelines often set by different institutions can be a major constraint for implementing these systems in large scale.
Under the clinical point of view, as future challenges, we highlight the recommendations to carry out more in-depth studies in the future, with larger samples and longer study times. Future work perspectives presented by the reviewed articles included mostly the refinement of current technologies and further evaluation and clinical validation of existing projects.”
- Line 401: One thing to consider is with regards to clinical validation. Importantly, it may be warranted to discuss EU and FDA approval for these as medical devices as opposed to using them simply as consumer technologies.
Answer: Thank you for your comment. In the Discussion section, we have reinforced the need of clinical validation, which was an aspect mentioned in several of the reviewed publications.
“Finally, published research lacks clinical validation and evaluation of the impact on health outcomes and well-being. In fact, Macdonald et al. (2021) [85] reported that diabetic populations remain optimistic about the role of technology in supporting foot monitoring but delivering evidence of wearable device efficacy in preventing foot ulcerations would improve trust and the likelihood of future adoption. International organizations such as Food and Drug Administration (FDA) and the European Commission present guidelines that deal with or encompass wearable devices used for medical purposes [86,87]. Considering that these devices work on multiple communication protocols, it is critical to establish their safety for human use before large scale implementation, thereby requiring the conduction of extensive clinical trials.”

Round 2
Reviewer 2 Report
I wish to thank the authors for making extensive edits to the manuscript, which is much improved.
While the authors indicate that the manuscript has been proofread there were still a number of grammatical errors, especially regarding the appropriate tense (ie plural vs singular). The authors should do an additional read through and correct errors as needed.